# Ethical Aspects of BCI Technology: What Is the State of the Art?

**Allen Coin** [ID], **Megan Mulder and Veljko Dubljević** *[ID]

Department of Philosophy and Religious Studies, NC State University, Raleigh, NC 27695-8103, USA;
allen_coin@ncsu.edu (A.C.); mkmulder@ncsu.edu (M.M.)

* Correspondence: veljko_dubljevic@ncsu.edu

**Abstract:** Brain–Computer Interface (BCI) technology is a promising research area in many domains. Brain activity can be interpreted through both invasive and non-invasive monitoring devices, allowing for novel, therapeutic solutions for individuals with disabilities and for other non-medical applications. However, a number of ethical issues have been identified from the use of BCI technology. In this paper, we review the academic discussion of the ethical implications of BCI technology in the last five years. We conclude that some emerging applications of BCI technology—including commercial ventures that seek to meld human intelligence with AI—present new and unique ethical concerns. Further, we seek to understand how academic literature on the topic of BCIs addresses these novel concerns. Similar to prior work, we use a limited sample to identify trends and areas of concern or debate among researchers and ethicists. From our analysis, we identify two key areas of BCI ethics that warrant further research: the physical and psychological effects of BCI technology. Additionally, questions of BCI policy have not yet become a frequent point of discussion in the relevant literature on BCI ethics, and we argue this should be addressed in future work. We provide guiding questions that will help ethicists and policy makers grapple with the most important issues associated with BCI technology.

**Keywords:** brain–computer interface (BCI); brain–machine interface (BMI); ethical; legal and social Issues (ELSI); neuroethics; narrative review

## 1. Introduction

Brain–Computer Interface (BCI) technology has been a promising area of research in recent decades, with advancements in the technology leading to a broadening of applications [1]. Researchers and clinicians are increasingly able to accurately interpret brain activity through both invasive (implanted) and non-invasive (outside the body) monitoring devices, allowing them to create better therapeutic solutions for patients suffering from disorders or diseases that inhibit their ability to interact with the world around them, e.g., patients suffering from the paralyzing locked-in syndrome who, with the use of a BCI device, are able to regain the ability to communicate. BCI technology is also being used for non-medical applications, such as gaming and human–technology interfaces. With this technology comes a number of ethical concerns that require consideration by all stakeholders involved (researchers, clinicians, patients and their families, etc.). Previous research into the ethics of BCI conducted by Burwell et al. [2] analyzed past publications that addressed ethical concerns associated with BCI technology. Burwell et al. identified common themes in the literature, including issues of responsibility for the consequences of BCI use; potential loss of autonomy, identity, and personhood as a result of BCI use; and security issues regarding the collection, analysis, and possible transmission of neural signals [2].

However, since Burwell et al. [2] conducted their original scoping review, there has been a rapid increase in the number of publications regarding BCI ethics. When conducting a literature search in 2020 using the same parameters that Burwell et al. used for their original search in 2016, we found that many additional and relevant articles [n = 34] had been published in the time since Burwell et al. conducted their study than had been published at any time before 2016, the last year included in the study [n = 42]. Additionally, there have been a number of advances in BCI technology in recent years, including commercial ventures that seek to utilize BCI in novel ways. One such example is the company Neuralink, founded by entrepreneur Elon Musk, which aims to achieve "a merger with artificial intelligence" [3]. There has been ample skepticism about Neuralink's goals and claims, with some referring to the company's public announcements and demonstrations as "neuroscience theater" [4]. Regardless of whether Neuralink's stated goals are feasible in the near-term future, the existence of commercial ventures like Neuralink in the BCI field certainly signals new areas of active development and may shed some light on where the technology could be heading.

One specific form of BCI development, Brain-to-Brain Interface (BBI), may lead to particularly novel social and ethical concerns. BBI technology combines BCI with Computer-to-Brain Interfaces (CBI) and, in newer work, multi-brain-to-brain interfaces—such as Jing et al.'s study [5]—real-time transfer of information between two subjects to each other has been demonstrated. Considering the rapid increase in publications about BCI ethics since 2016 and recent advances in the technology, a review of the state of the art of the ethical discussion of BCI is warranted. With these developments in mind, we review the academic discussion of the ethical implications of BCI in the last five years. Through this type of systematized qualitative analysis [6], we hope to provide a nuanced perspective on the complicated ethical implications of this technology and directions for its responsible development and use that will ensure it advances in an ethically sound manner.

In the following Background section, we will provide a detailed summary of Burwell and colleagues' findings before discussing our own research in the sections Materials and Methods, Results, Discussion, and Conclusion.

## 2. Background

In 2017, Burwell et al. published the first scoping review analyzing themes surrounding ethical issues in BCI. PubMed was used to find these articles using advanced searches combining various relevant terms (including Medical Subject Heading/MeSH terms) denoting "brain-computer interfaces" with keywords pertaining to ethics in general. From these articles, they identified narrower ethical concerns, such as "personhood," "stigma," and "autonomy." From an initial yield of around 100 documents, Burwell and colleagues selected 42 articles that met their inclusion criteria for this review. They included only papers that were written in English, presented discussions or empirical findings of the ethics of BCI, referred to human subjects, and considered BCI as technology that records data directly from the brain to a computer output. To provide direction for their study, Burwell et al. also consulted with four experts in the related research areas of clinical medicine, biomedical engineering, bioethics, and end-user perspectives.

After coding the selected articles, they found that most articles discussed more than one ethical issue, with the minority of the articles in question being empirical papers. The most frequently mentioned ethical issues included User Safety [57.1%, n = 24], Justice [47.6%, n = 20], Privacy and Security [45.2%, n = 19], and Balance of Risks and Benefits [45.2%, n = 19]. The authors focused primarily on the ethical issues of (i) User Safety, (ii) Humanity/Personhood, (iii) Stigma and Normality, (iv) Autonomy, (v) Responsibility, (vi) Research Ethics and Informed Consent, (vii) Privacy and Security, and (viii) Justice. Burwell and colleagues also briefly mentioned the less-cited concerns of military applications, enhancement and transhumanism, general societal impacts, and BCI technology regulation.

The issue of User Safety focused on potential direct physical harms to the user if the technology was to fail, an example being crossing the street with prosthetics when the BCI gives out. Ethicists that

conducted research on this topic also discussed the unknown side effects of BCI, including the mental and physical toll of learning to use the technology.

The themes of Humanity and Personhood concerned whether the BCI would become part of the user's "body schema," and there was little uniformity in opinion on this issue among researchers. Some emphasized the fact that humans are already intricately linked to technology and that a BCI is no different, while one researcher went so far as to say we could evolve from "*homo sapiens*" to "*homo sapiens technologicus*" with technologies such as these [7]. BCI users interviewed in several qualitative empirical studies were discomforted by this possibility and distanced themselves from the idea of cyborgization.

The themes of Stigma and Normality addressed the BCI user's ability to influence or be influenced by the social stigma of disability. It is a possibility that individuals who feel stigmatized by a disability will consent to a BCI to counteract that. On the other hand, however, having a BCI could become stigmatized instead of the disability itself.

The issue of Responsibility was also salient in ethical discussions. If a negative action were to be carried out by someone using a BCI, would it be the user's fault or the fault of the technology? Many researchers claim that our legal system is not yet equipped to deal with this situation.

The issue of Autonomy refers to one's ability to act independently and of their own volition. If an action is only possible because of BCI technology, is that damaging to one's autonomy? On the other hand, autonomy may be increased on the basis that individuals living with BCI technologies may be able to do things on their own that once required constant supervision and assistance by a caregiver.

The nature of a BCI sending brain signals directly to a computer raises the possibility of hacking. This issue is connected to concerns regarding Privacy and Security. BCIs could also be used to extract information from a person such as their current mental state or truthfulness, or even be utilized to cause harm to the user or someone else.

Another major ethical issue identified was that of Research Ethics and Informed Consent. Many potential BCI users are people living in a locked-in state (almost complete paralysis), or with other conditions that limit their ability to give consent. Even if that is not the case, a severe disability may press an individual to consent to using BCI technology out of desperation without fully considering the risks.

The final main ethical concern discussed by Burwell et al. was Justice. Many researchers worry that the perspectives of those using BCIs are not fully considered in the research. There are also many questions pertaining to the fairness of providing only limited access to BCI technology. For instance, when experimental BCI studies are completed, do the participants get to keep the BCI? Should BCIs become widely available, there is also the concern that enhancement could potentially exacerbate existing inequalities between individuals and social strata, e.g., a scenario in which a BCI technology that provides cognitive or physical enhancement [8] is only available to the wealthy.

While it's not surprising to see many potential ethical issues and questions arising from use of a novel technology, what is surprising is the lack of suggestions to resolve them. One hope that Burwell and colleagues had for their scoping review was to facilitate informed recommendations for future research and development of neurotechnology. Our research, building upon the groundwork of Burwell and colleagues, sheds light on how the discourse on BCI ethics has evolved in the years since.

## 3. Materials and Methods

Building on prior work by Burwell et al., in April of 2020 we conducted a search using PubMed and PhilPapers in order to track academic discussion of this technology since 2016. The search terms were selected to mirror the search done by Burwell's original work on the subject. The following search queries were used.

PubMed: (("brain computer interface" OR "BCI" OR "brain machine interface" OR "Brain-computer Interfaces"[Mesh]) AND (("personhood" OR "Personhood"[Mesh]) OR "cyborg" OR "identity" OR ("autonomy" OR "Personal autonomy"[Mesh]) OR ("liability" OR "Liability,

Legal"[Mesh]) OR "responsibility" OR ("stigma" OR "Social stigma"[Mesh]) OR ("consent" OR "Informed Consent"[Mesh]) OR ("privacy" OR "Privacy"[Mesh]) OR ("justice" OR "Social Justice"[Mesh]))).

PhilPapers: ((brain-computer-interface|bci|brain-machine-interface)&(personhood|cyborg| identity|autonomy|legal|liability|responsibility|stigma|consent|privacy|justice)).

We sought to understand how the academic literature on the topic since 2016 addresses unique, new ethical concerns presented by emerging applications of BCI. While Burwell et al. identified 42 relevant articles published before 2016, our slightly modified search in 2020 using a similar methodology and exclusion/inclusion criteria yielded almost as many relevant articles discussing the ethics of BCI [n = 34] published since 2016. We used a limited, randomly selected sample [20.6%, n = 7] from the pool of 34 articles to identify trends and areas of concern or debate among researchers and ethicists, especially regarding topics like autonomy, privacy and security, and informed consent/research ethics. Additionally, while Burwell and colleagues only considered articles about BCI ethics specifying human subjects, we expanded our search and inclusion/exclusion criteria to include applications involving animals and other subjects, such as brain organoids [9].

Based on the abductive inference approach to qualitative research [10], we used the thematic framework developed by Burwell and colleagues to identify and map the overarching themes of ethical issues posed by BCIs (see Figure 1). The map identifies eight specific ethical concerns that define the conceptual space of the ethics of BCI as a field of research. Only one of the ethical concerns refers to physical factors specifically: User Safety. Two are explicitly about psychological factors: Humanity/Personhood and Autonomy; while the remaining five focus on social factors: Stigma and Normality, Responsibility and Regulation, Research Ethics and Informed Consent, Privacy and Security, and Justice.

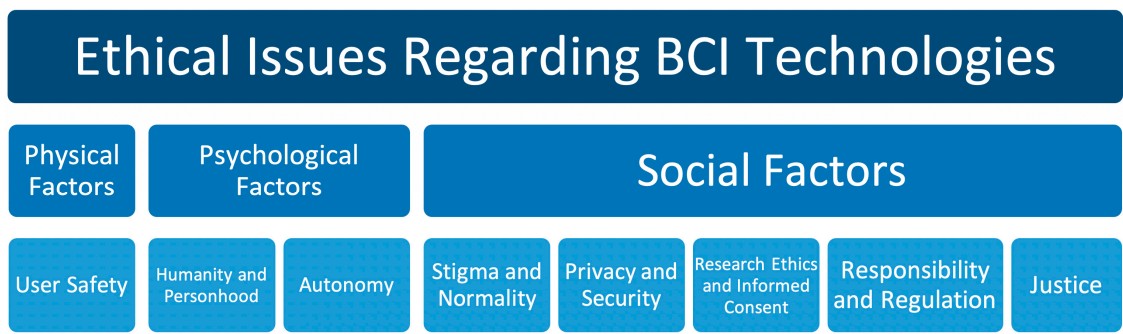

**Figure 1.** Overarching Themes in Brain–Computer Interface (BCI) Ethics.

## 4. Results

In the updated sample, we were able to locate mentions of all eight categories, with some ethical issues, such as Autonomy, being mentioned much more frequently [71.4%, n = 5] than others. While most articles discussed benefits in terms of the increases in autonomy and independence gained from using a BCI [11–14], the potential for autonomy to be compromised was also discussed. For example, Hildt [15] mentions the possibility of taking the information gained from BCI—or in this case, Brain-to-Brain Interface (BBI)—from the individual and using it without their consent or knowledge:

> "Participants in BBI networks depend heavily on other network members and the input they provide. The role of recipients is to rely on the inputs received, to find out who are the most reliable senders, and to make decisions based on the inputs and past experiences. In this, a lot of uncertainty and guessing will be involved, especially as it will often be unclear where the input or information originally came from. For recipients in brain networks, individual or autonomous decision-making seems very difficult if not almost impossible" [15] (p. 3).

Another frequently [57.1%, n = 4] discussed ethical issue was Humanity and Personhood, since BCIs could impact one's sense of self. In one specific study of BCI technology used in patients with epilepsy, there was a variety of resulting perspectives on sense of self, with some individuals saying that it made them feel more confident and independent, while others felt like they were not themselves anymore, with one patient expressing that the BCI was an " . . . extension of herself and fused with part of her body . . . " [11] (p. 90). Other articles more generally discussed the possibility of sense of self changing and the ways BCI technology could contribute to this. Sample and colleagues categorize three ways in which one's sense of self and identity could change: altering the users' interpersonal and communicative life, altering their connection to legal capacity, and by way of language associated with societal expectations of disability [14] (p. 2). Müller and Rotter argue that BCI technology constitutes a fusion of human and machine, stating that "the direct implantation of silicon into the brain constitutes an entirely new form of mechanization of the self . . . [T]he new union of man and machine is bound to confront us with entirely new challenges as well" [13] (p. 4).

Research Ethics and Informed Consent was also a frequently [57.1%, n = 4] mentioned issue. The main consensus among the ethicists that discussed this was that it is very important to obtain informed consent and make sure that the subjects are aware of all possible implications of BCI technology before consenting to use it. Additionally, some ethicists warned against the possibility of exploiting potentially vulnerable BCI research subjects. As Klein and Higger note: "[t]he inability to communicate a desire to participate or decline participation in a research trial—when the capacity to form and maintain that desire is otherwise intact—undermines the practice of informed consent. Individuals cannot give an informed consent for research if their autonomous choices cannot be understood by others" [12] (p. 661).

User Safety was discussed as often as research ethics in the sample [57.1%, n = 4], with both psychological and physical harm being mentioned and explained as serious possibilities that need to be considered [13–15]. One article discussed the impacts of harm on the results of a BCI study, stressing the importance of stopping a clinical trial if the risks to the individual participants begin to outweigh the potential benefits to science [12].

Issues of User Safety led to discussions of Responsibility and Regulation when using BCI technology. While the term "regulation" was mentioned in several articles [57.1%, n = 4], only one went into significant detail about regulation in regard to BCIs specifically, discussing the issue of a "right to brain privacy," which can be understood in similar terms to existing privacy legislation, such as General Data Protection Regulation (GDPR) in the European Union or the Health Insurance Portability and Privacy Act (HIPAA) in the United States, to regulate the information gathered in BCI use [15] (p. 2). This was also the only time regulation was used in a legal sense for the technology, as opposed to regulating who it should be used for [11]. Responsibility was also mentioned multiple times [71.4%, n = 5], but again, only one article went into detail about who would be responsible for potentially dangerous or illegal BCI technology uses [16]. One discussed the "responsibility" of the research being divided among members of the research team [12], which was not Burwell et al.'s original meaning of the category Responsibility.

*Privacy and Security* concerns were mentioned somewhat less frequently [42.8%, n = 3] but still discussed in depth. Three articles [13,15,17] talked about the risks of extracting private information from people's brains and using it without their knowledge or consent, which is a significant concern for BCI technologies. Müller & Rotter connected this issue to User Safety, arguing that the increased fidelity of BCI data yields inherently more sensitive data, and that the "impact of an unintended manipulation of such brain data, or of the control policy applied to them, could be potentially harmful to the patient or his/her environment" [13] (p. 4).

*Justice* was also mentioned infrequently [28.6%, n = 2], with the main idea being inequality and injustice within the research. These discussions often related back to the aforementioned questions of when the trials would end and if the participants would get to subsequently keep the BCI technologies [12].

The final two social factors mentioned were Stigma and Normality [28.6%, n = 2] and Societal Implications [28.6%, n = 2]. Stigma was mainly discussed from the perspective of the device itself having a negative stigma around it, and the device itself being what is stigmatizing about the individual [12]. However, it was also mentioned that perhaps universalizing the technology instead of only targeting it toward a group that is considered "disabled" could reduce or eliminate stigma [14]. Societal Implications were discussed from several standpoints. One take was the possibility of BCI being used as a sort of social network instead of just for therapy for disabled individuals [15]. Another position was that—since society tends to universalize technology so that it is used by nearly everyone and hard to function without, e.g., cellphones—the use of BCI technology may become a "... precondition to the realization of personhood" [14]. This article also discussed the potential for BCI technology to reshape how society perceives disability.

Similar to Burwell et al.'s findings, Military Use and Enhancement were both mentioned rarely [14.3%, n = 1], and neither category was discussed any further than using it as an example for potential BCI use [15]. For the more prominent categories, the distribution can be seen in Table 1.

**Table 1.** The distribution of overarching themes of BCI ethics.

|  | Burwell et al.'s Distribution out of 42 Selected Papers (2016) | Our Distribution out of a Sample of 7 Selected Papers (2020) |
|---|---|---|
| *User Safety* | 57.1% | 57.1% |
| *Humanity and Personhood* | 35.7% | 57.1% |
| *Autonomy* | 28.6% | 71.4% |
| *Stigma and Normality* | 26.2% | 28.6% |
| *Privacy and security* | 45.2% | 42.8% |
| *Research Ethics and Informed Consent* | 33.3% | 57.1% |
| *Responsibility and Regulation* | 31.0% | 71.4% |
| *Justice* | 47.6% | 28.6% |

## 5. Discussion

While there have been notable advancements in BCI and BBI technology and the body of literature of the ethical aspects of BCI technology has grown substantially since the original publication of Burwell and colleagues' research, our findings suggest that the original taxonomy developed by Burwell and colleagues remains a useful framework for understanding the body of literature specifically on the social factors of the ethics of BCI. We can use this taxonomy, with some slight modifications, which we outline below, to understand how the body of literature on the ethics of BCI is grappling with ethical issues arising from the applications of this rapidly advancing technology. Articles published since 2016 still mostly conform to the taxonomy and can be categorized using it in future iterations of the scoping review methodology.

There are, however, some areas within the growing body of literature on BCI ethics that have arisen since the original research was published that need to be incorporated into the taxonomy. We recommend the following modifications to the conceptual mapping outlined in Figure 1. First, expanding the discussion of the physical (e.g., harms to test animals) and psychological (e.g., radical psychological distress) effects of BCI technology. The publicly available information on commercial BCI endeavors (such as Neuralink) frequently mentions experiments with increasingly complex and even sentient animals, such as Neuralink's demonstration of their technology on live pigs [4]. The lack of ethical scrutiny of these studies is an essential cause for concern [18]. Thus, ethical discussions should be expanded to include public awareness of private industry research into BCI using animals. Secondly, while the risks of physical harm from BCI are fairly well-understood and covered in the literature, further research is needed to understand emerging psychological factors in BCI ethics, examining how human–AI intelligence symbiosis, brain-to-brain networking, and other novel applications of the technology [2] may affect psychological wellbeing in humans. For instance, in the interview study by

Gilbert and colleagues, one patient mentioned that "she was unable to manage the information load returned by the device," which led to radical psychological distress [11] (p. 91).

Going forward, it is imperative to expand on the connection between ethics and policy in discussions of BCI technology and conduct more empirical studies that will help separate non-urgent policy concerns, which are based on not-yet attained effects of BCI, from the more urgent concerns based on the current state of science in regards to BCI technology. In this, we echo Voarino and colleagues [19] in stating that we must advance the discussion from merely mapping ethical issues, into an informed debate that explains which ethical concerns are high priority, which issues are moderately important, and what constitutes a low priority discussion of possible future developments.

That said, it is important to make sure that the ethics literature keeps pace with engineering advances and that policy does not lag behind. In that vein, following Dubljević [20], we propose that the key ethical question for future work on BCI ethics is:

> What would be the most legitimate public policies for regulating the development and use of various BCI neurotechnologies by healthy adults in a reasonably just, though not perfect, democratic society?

Additionally, we need to distinguish between ethical questions regarding BCI technology that ethicists and social scientists can answer for policy makers and those that cannot be resolved even with extensive research funding [21]. Therefore, following Dubljević and colleagues [22], we posit that these four additional questions need to be answered to ensure that discussions of BCI technology are realistic:

1. What are the criteria for assessing the relevance of BCI cases to be discussed?
2. What are the relevant policy options for targeted regulation (e.g., of research, manufacture, use)?
3. What are the relevant external considerations for policy options (e.g., international treaties)?
4. What are the foreseeable future challenges that public policy might have to contend with?

By providing answers to such questions (and alternate or additional guiding questions proposed by others), ethicists can systematically analyze and rank issues in BCI technology based on an as-yet to be determined measure of importance to society. While we have not completed such analyses yet, we do provide a blueprint above, based on conceptual mapping and newly emerging evidence, of how this can be done.

## 6. Conclusions

This article builds on, and updates, previous research conducted by Burwell and colleagues to review relevant literature published since 2016 on the ethics of BCI. Although that article is now somewhat outdated in terms of specific references to and details from the relevant literature, the thematic framework and the map we created—with the eight specific categories that it provides—and the nuanced discussion of overarching social factors have withstood the test of time and remain a valuable tool to scope BCI ethics as an area of research. A growing body of literature focuses on each of the eight categories, contributing to further clarification of existing problems. BCI ethics is still in its early stages, and more work needs to be done to provide solutions for how these social and ethical issues should be addressed.

Despite seeing the significance of these eight categories continue into more recent research, it is worth noting that we found that the distribution of the eight categories was different in recent years, compared with the distribution previously identified by Burwell and colleagues in the literature published before 2016. For instance, among our sample of articles, we found that Autonomy was mentioned most frequently [71.4%, n = 5] along with Responsibility and Regulation [71.4%, n = 5], with Research Ethics, User Safety and Humanity and Personhood each discussed in 4 out of 7 [57.1%] of the articles in the sample. However, despite Responsibility and Regulation being mentioned in five out of the seven papers, it was only discussed at length in one. None of these categories were among

Burwell and colleagues' top four most frequently mentioned (see Table 1). It seems that while the eight issues mapped are still ethically significant with regards to BCI research, the emphasis among them may be shifting toward concerns of psychological impact.

On that note, psychological effects (e.g., radical psychological distress) need to be carefully scrutinized in future research on BCI ethics. Additionally, one aspect that was not explicitly captured in the original thematic framework or the map we reconstructed from it is physical harm to animals used in BCI experimentation [18]. Finally, more detailed proposals for BCI policy have not yet become a frequent point of discussion in the relevant literature on BCI ethics, and this should be addressed in future work. We have provided guiding questions that will help ethicists and policy makers grapple with the most important issues first.

**Author Contributions:** A.C. contributed with Conceptualization, Data curation, Formal analysis, Methodology, Project administration, Writing—original draft, Writing—review & editing. M.M. contributed with Data curation, Writing—original draft, Writing—review & editing. V.D. contributed with Conceptualization, Formal analysis, Methodology, Project administration, Supervision, Writing—original draft, Writing—review and editing. All authors have read and agreed to the published version of the manuscript.

**Funding:** This research received no external funding.

**Acknowledgments:** The authors would like to thank for their valuable discussion and feedback the members of the NeuroComputational Ethics Research Group at NC State University—in alphabetical order, Elizabeth Eskander, Anirudh Nair, Sean Noble, and Abigail Presley. Additionally, the authors would like to thank Joshua Myers (NC State University) for his assistance with the early stages of this paper.

**Conflicts of Interest:** The authors declare no conflict of interest.

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
