# Peer review of "Ethical Aspects of BCI Technology: What Is the State of the Art?"

_philosophies, doi:10.3390/philosophies5040031_

Round 1

Reviewer 1 Report

General impression

In general I enjoyed reading this paper. It is written in a clear matter, and it is an important topic.

I have some doubts about the quantitative approach of the paper. I feel that the content of the ethical arguments become a bit ‘flat’ this way, especially because the authors refer to categories, rather than to ethical statements. The amount of papers is a bit low for such a quantitative approach, and I don’t really see how this quantification help them in determining which issues have priority.

Personally, I like this paper by Strech and colleagues about how to do quantative reviews of ethical issues:

Strech, D., Synofzik, M., & Marckmann, G. (2008). Systematic reviews of empirical bioethics. Journal of Medical Ethics, 34(6), 472–477.

I would like to recommend the authors to read it for future work in this direction.

But it would be a bit far reaching to ask the authors to change their methodology for this paper.

I have some small clarification issue, and some larger issues.

Small clarification

I have a question about this section in the methodology section:

‘We used a limited sample [20.6%, n=7] to identify trends and areas of concern or debate among researchers and ethicists, especially regarding topics like autonomy, privacy and security, and informed consent/research ethics.’

How is this limited sample selected? And why focus especially on these topics?

What is meant by the limited sample? Is it a sample from the articles they found, or did they do a refined search on the mentioned topics?

BTW: I really liked the figure!

Suggestion: I am not sure if I would name it ‘psychological factors’. Maybe ‘Personhood’ or ‘Personal factors’. And Stigma could both be a psychological/personal factor as a social one.

I don’t recommend necessarily that it is adjusted, just a food for thoughts things. Every categorization has up- and downsides.

Larger issues

What is discussed now in the discussion section, especially on how the model should be expanded with new themes, should be part of the result section.

I don’t know what the word limit of the journal is, but I would like to see the section on ‘new themes’ to be expanded a bit. To more clearly describe the content.

Normally in reviews, there is an overview of studies included, I would like to see that added. And maybe also a table with the studies from the original review of Burwell.

In think the section from line 271 on, is very interesting, however, it does not become clear to me how it follows from the review you did. It’s too quick and jumpy now, and I would like to see more elaboration on it.

For example: which issues cannot be resolved? Why is the question you raise the key question? Where are the four additional questions coming from? And how do they relate to the result presented I your paper?

For readers who don’t know Dubljević previous paper, it is not really clear what you mean with that section.

So my suggestion would be to add the section on new themes to your results section, and elaborate on it, and expand the questions raised in the final part of the discussion.

Reviewer 2 Report

General comments

This paper reviews the literature from the last five years on ethical aspects of BCIs. It is essentially an update of Burwell, Sample & Racine (2017). The paper is clearly and well written, and it provides a useful update on the ethics literature on BCIs. The methods section lacks detail about the selection of literature and other aspects of the methods, referring to “same search terms” and “using similar methodology” as Burwell et al., as well as “used the thematic framework developed by Burwell”. I suppose the reader can look this up in Burwell et al., but maybe the authors could at least indicate main differences if any, apart from the expansion of the scope beyond human subjects. The selective sample of seven papers requires explanation.

Specific comments

p. 2, line 95: The paragraph appears to be a summary of the discussion of Burwell et al. about Humanity and Personhood. If that is the aim, then the formulation “Some compared [BCI] to cochlear implants ... implying that it does not affect personhood while others went so far as ...” is somewhat inaccurate. The one side of the contrast is about how our bodies are already intricately linked with tools and technology, seemingly suggesting that this linking is unproblematic, but note that neither cochlear implants nor personhood is mentioned in this part of the original section, the contrast seems to be about “humanity”. “While others” appears to be just Zehr 2015 (Burwell et al. ref. [34]).

p. 3, line 103: Is there a reason why the discussion about autonomy is not summarised but all the other topics are? It is a bit odd given that autonomy is mentioned more often than any other issue.

p. 3, line 136: Why did the authors use a limited sample of seven articles out of 34 and how was this sample selected? The small sample raises the issue of whether the sample is representative for the wider literature, and the lack of any indication of how the sample was selected makes that issue more problematic.

p. 6, lines 263–264: I think it would be appropriate to reference here Burwell et al. again, since this is a formulation of their two recommendations for neuroethics research (more work on policy/practical guidelines and more empirical work).
